# BHLHE40, a third transcription factor required for insulin induction of SREBP-1c mRNA in rodent liver

Jing Tian[1], Jiaxi Wu[2], Xiang Chen[2], Tong Guo[1], Zhijian J Chen[2], Joseph L Goldstein[1]*, Michael S Brown[1]*

[1]Department of Molecular Genetics, University of Texas Southwestern Medical Center, Dallas, United States; [2]Department of Molecular Biology and Howard Hughes Medical Institute, University of Texas Southwestern Medical Center, Dallas, United States

**Abstract** In obesity, elevated insulin causes fatty liver by activating the gene encoding SREBP-1c, a transcription factor that enhances fatty acid synthesis. Two transcription factors, LXRα and C/EBPβ, are necessary but not sufficient for insulin induction of hepatic SREBP-1c mRNA. Here, we show that a third transcription factor, BHLHE40, is required. Immunoprecipitation revealed that BHLHE40 binds to C/EBPβ and LXRα in livers of rats that had fasted and then refed. Hepatic BHLHE40 mRNA rises rapidly when fasted rats are refed and when rat hepatocytes are incubated with insulin. Preventing this rise by gene knockout in mice or siRNAs in hepatocytes reduces the insulin-induced rise in SREBP-1c mRNA. Although BHLHE40 is necessary for insulin induction of SREBP-1c, it is not sufficient as demonstrated by failure of lentiviral BHLHE40 overexpression to increase hepatocyte SREBP-1c mRNA in the absence of insulin. Thus, an additional event is required for insulin to increase SREBP-1c mRNA.
DOI: https://doi.org/10.7554/eLife.36826.001

*For correspondence:
joe.goldstein@utsouthwestern.edu (JLG);
mike.brown@utsouthwestern.edu (MSB)

Competing interests: The authors declare that no competing interests exist.

## Introduction

Sterol regulatory element-binding protein-1c (SREBP-1c) is a transcription factor that activates fatty acid synthesis and thereby induces fatty liver in obese rodents (*Horton et al., 2002*; *Moon et al., 2012*). The level of SREBP-1c in liver is controlled by insulin (*Shimomura et al., 1999*; *Foufelle and Ferré, 2002*). When animals fast and plasma insulin is low, the amount of SREBP-1c mRNA in liver is low. SREBP-1c mRNA levels rise 8–30 fold when insulin secretion is stimulated by refeeding a high carbohydrate diet to rodents that were previously fasted (*Horton et al., 1998*; *Liang et al., 2002*; *Li et al., 2010*). The resultant increase in nuclear SREBP-1c leads to the characteristic insulin-induced increase in fatty acid and triglyceride synthesis. Under conditions of obesity and insulin resistance, insulin loses its ability to suppress gluconeogenic mRNAs in liver, yet the hormone continues to stimulate SREBP-1c production and fatty acid synthesis in that organ (*Brown and Goldstein, 2008*; *Titchenell et al., 2016*). This phenomenon has been termed 'selective insulin resistance' and it leads to a pathologic state in which the liver produces glucose and fatty acids simultaneously, a phenomenon that results in hyperglycemia, hypertriglyceridemia, fatty liver and cirrhosis (*Brown and Goldstein, 2008*). Solving this problem will require an understanding of the mechanism by which insulin stimulates production of SREBP-1c mRNA. The current experiments were designed to increase this understanding.

Previous studies identified liver X receptors (LXRs) and C/EBPβ as transcription factors that are required for insulin stimulation of SREBP-1c mRNA production (*Chen et al., 2004*; *Tian et al., 2016*; *Millward et al., 2007*). Immunoprecipitation studies revealed that LXRα and C/EBPβ form a complex

that binds to a region of the SREBP-1c promoter that contains two canonical LXR-binding sites (*Tian et al., 2016*). These sites were shown previously to be required for insulin induction of SREBP-1c transcription (*Chen et al., 2004*). Deficiency of LXR or C/EBPβ prevents insulin induction of SREBP-1c mRNA and fatty acid synthesis (*Kalaany et al., 2005*; *Tian et al., 2016*). Chromatin immunoprecipitation assays revealed that insulin does not activate the LXRα-C/EBPβ complex directly. Indeed, the complex is bound to the SREBP-1c promoter in the absence as well as the presence of insulin (*Tian et al., 2016*). This finding suggests that the liver must contain another protein that activates the pre-existing LXRα-C/EBPβ complex when insulin is present.

The current study was designed to identify an insulin-regulated factor that acts together with the LXRα-C/EBPβ complex to increase SREBP-1c transcription in the presence of insulin. Through quantitative mass spectrometry, we identified a single phosphorylated protein that was co-immunoprecipitated with LXRα and C/EBPβ in nuclear extracts of livers from refed but not fasted rats. The protein, which we designate as BHLHE40 (Basic Helix-Loop-Helix Family Member E40), has been observed by others in many organs and contexts and has received several other names, including SHARP-2, STRA13, and DEC1. We use the name BHLHE40 because that is the name given to its gene (http://www.genecards.org/cgi-bin/carddisp.pl?gene=BHLHE40). Previous studies showed that insulin induces production of BHLHE40/SHARP-2 in rat liver (*Kanai et al., 2017*; *Teboul et al., 1995*). However, this induction was not postulated to be involved in SREBP-1c gene activation. Here, we provide evidence that induction of BHLHE40 is a required step in insulin-mediated activation of the SREBP-1c gene, and therefore it is required for insulin-mediated stimulation of fatty acid synthesis.

## Results

To search for a factor that works together with LXRα and C/EBPβ to mediate insulin induction of SREBP-1c gene transcription, we sought to identify a nuclear protein that was enriched in livers of fed rats and that bound to both of these transcription factors. As shown in *Figure 1A*, we prepared nuclear extracts from livers of rats that were either fasted for 48 hr to reduce plasma insulin or fasted for 48 hr and then refed a high-carbohydrate diet for 6 hr in order to induce insulin secretion. Separate aliquots of the nuclear proteins were subjected to immunoprecipitation with either anti-LXRα or anti-C/EBPβ. The precipitated proteins were subjected to SDS-PAGE and identified by mass spectrometry. Using label-free quantification, we identified a total of 32 proteins that were enriched by at least threefold in the extracts from refed rats as compared with fasted rats and that were also precipitated by both anti-LXRα and anti-C/EBPβ (*Figure 1B*, shown in red). Inasmuch as insulin signaling is mediated by a cascade of phosphorylation reactions, we subjected the immunoprecipitates to LC-MS/MS analysis to determine whether any of the 32 refeeding-induced proteins was phosphorylated. Only one of the 32 proteins fulfilled this criterion, namely, BHLHE40, which was observed to be phosphorylated on serine 383 (*Figure 1B–D*). The mass spectroscopy data indicated that BHLHE40 protein was increased by 18-fold in the anti-LXRα immunoprecipitates from refed rats as compared with fasted rats. The increase was 9.8-fold in the anti-C/EBPβ immunoprecipitates (*Figure 1C*).

*Figure 2* shows an experiment in which we used real-time PCR to quantify mRNAs in liver extracts from fasted and refed rats. As shown previously (*Liang et al., 2002*), refeeding increased the SREBP-1c mRNA by 18-fold (*Figure 2B*). Refeeding increased the BHLHE40 mRNA by 8.4-fold (*Figure 2A*). The LXRα mRNA rose less than twofold (*Figure 2C*) and the C/EBPβ mRNA was not increased by refeeding (*Figure 2D*).

To confirm the co-immunoprecipitation of BHLHE40 with LXRα and C/EBPβ, we prepared liver nuclear extracts from fasted and refed rats (*Figure 3*). The extracts were split into four equal portions. One portion was reserved as the input fraction, and the other three were incubated with agarose beads coated with anti-IgG, anti-LXRα or anti-C/EBPβ. Aliquots containing 30% of the input fraction and all of the proteins eluted from the beads were subjected to SDS-PAGE and immunoblotted with antibodies directed against LXRα, C/EBPβ, BHLHE40, SREBP-1c and CREB. SREBP-1c and CREB, like BHLHE40, are basic-helix-loop-helix transcription factors. They were included as controls for nonspecific interaction. In the extracts from refed rats, BHLHE40 was precipitated by anti-LXRα and anti-C/EBPβ (*lanes 6 and 8*), but not by the control anti-IgG (*lanes 3 and 4*). Neither SREBP-1c nor CREB was precipitated by anti-LXRα or anti-C/EBPβ (*lanes 5–8*). These data show that anti-LXRα and anti-C/EBPβ can precipitate BHLHE40 from refed rats, consistent with the formation of a LXRα-C/EBPβ-BHLHE40 complex.

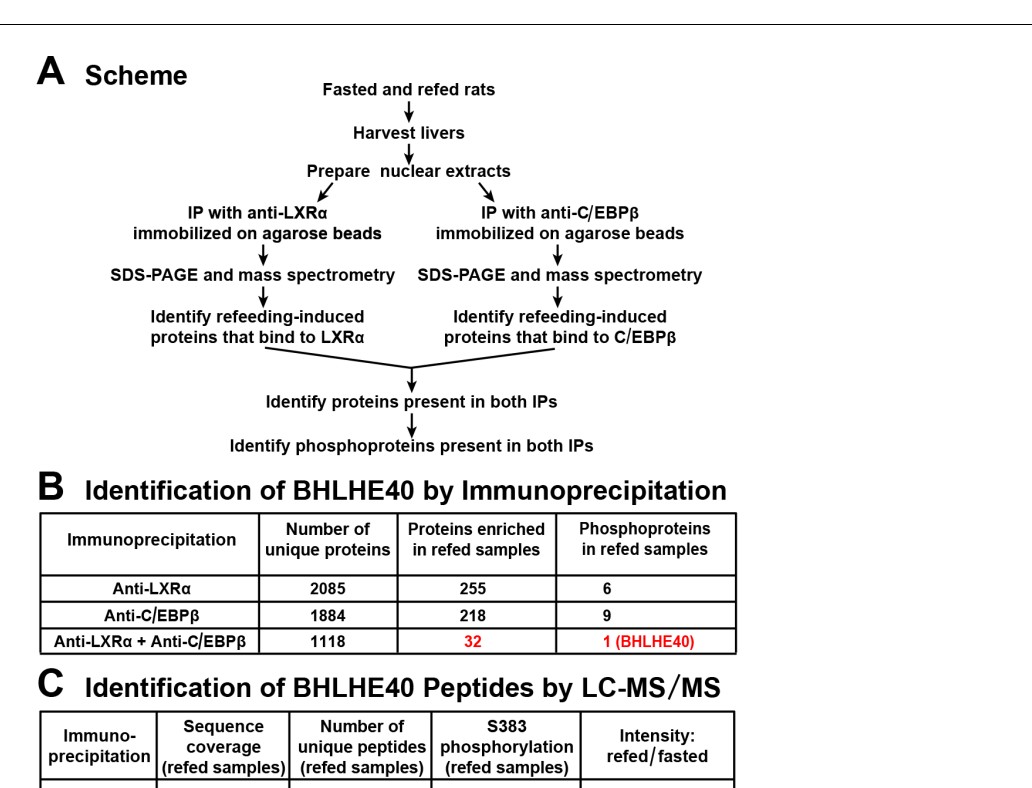

## B  Identification of BHLHE40 by Immunoprecipitation

| Immunoprecipitation | Number of unique proteins | Proteins enriched in refed samples | Phosphoproteins in refed samples |
|---|---|---|---|
| Anti-LXRα | 2085 | 255 | 6 |
| Anti-C/EBPβ | 1884 | 218 | 9 |
| Anti-LXRα + Anti-C/EBPβ | 1118 | 32 | 1 (BHLHE40) |

## C  Identification of BHLHE40 Peptides by LC-MS/MS

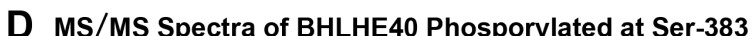

| Immuno-precipitation | Sequence coverage (refed samples) | Number of unique peptides (refed samples) | S383 phosphorylation (refed samples) | Intensity: refed/fasted |
|---|---|---|---|---|
| Anti-LXRα | 28% | 11 | yes | 18.0 |
| Anti-C/EBPβ | 39% | 14 | yes | 9.8 |

## D  MS/MS Spectra of BHLHE40 Phosporylated at Ser-383

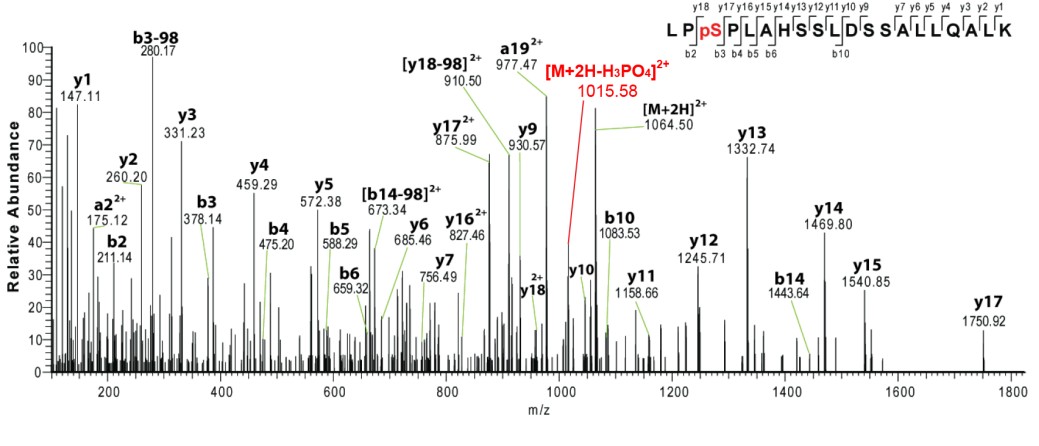

**Figure 1.** Co-immunoprecipitation of BHLHE40 phosphorylated at serine-383 (S383). (**A**) Protocol for identifying the phosphoproteins that are induced by feeding and interact with both LXRα and C/EBPβ. Rats were either fasted for 48 hr, or fasted for 48 hr and then refed with a high-carbohydrate diet for 6 hr. Nuclear extracts were prepared from livers and subjected to immunoprecipitation with anti-LXRα or with anti-C/EBPβ, after which the immunoprecipitates were subjected to 10% SDS-PAGE and analyzed by mass spectrometry. Phosphoproteins that were enriched (threefold or greater) in refed samples and co-immunoprecipitated with both anti-LXRα and anti-C/EBPβ were identified. (**B**) LC-MS/MS analysis of immunoprecipitated proteins. (**C**) Identification of BHLHE40 peptides. The intensity ratio (refed/fasted) was calculated using the summed intensity of all peptides in the protein. (**D**) MS/MS spectra data of BHLHE40 phosphorylated at Ser-383. LC-MS/MS sample preparation and analysis are described in Materials and methods.

DOI: https://doi.org/10.7554/eLife.36826.002

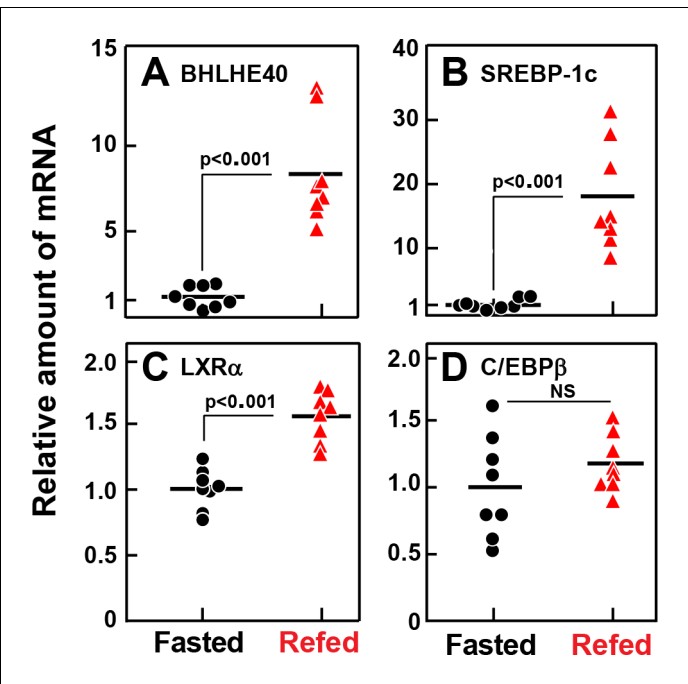

**Figure 2.** Relative amounts of mRNA for BHLHE40 (**A**), SREBP-1c (**B**), LXRα (**C**), and C/EBPβ (**D**), in livers of rats subjected to fasting and refeeding. Male rats (age 2–3 months) were either fasted for 48 hr, or fasted for 48 hr and then refed with a high-carbohydrate diet for 6 hr. Total liver RNA from eight rats in each group was subjected to quantitative RT-PCR. Each black circle (fasted rats) or red triangle (refed rats) represents an individual animal. Each value in the refed group represents the amount of mRNA relative to the mean amount in the fasted group, which is arbitrarily defined as 1.0. Mean Ct values for BHLHE40, SREBP-1c, LXRα, and C/EBPβ in the fasted groups were 24.9, 23.9, 24.9, and 24.4, respectively. NS, not significant. p-Values calculated using Student's t-test.

DOI: https://doi.org/10.7554/eLife.36826.003

To study the binding of the LXRα-C/EBPβ-BHLHE40 complex to the SREBP-1c enhancer/promoter region, we performed ChIP assays on DNA from livers of fasted and refed rats (*Figure 4*). Slices of rat liver were incubated with formaldehyde to cross-link proteins to DNA. The tissue was disrupted, and the DNA was sheared by sonication. After preclearance with a protein A agarose/salmon sperm DNA resin, the supernatant was incubated with various antibodies, and the antibody-bound protein/DNA complexes were precipitated with protein A agarose. The precipitated DNA was subjected to PCR with primer pair 1 (*Figure 4A*), which amplifies the SREBP-1c promoter/enhancer region that includes two LXR-binding sites (*Chen et al., 2004*) or with primer pair 2, which amplifies a region 5 kb upstream. In liver samples amplified by primer pair 1, anti-BHLHE40 brought down more DNA from the fed than the fasted livers (*Figure 4B*, *lanes 6 and 5*). Anti-LXRα and anti-C/EBPβ brought down the same LXRE region in samples from both fasted and fed rats (*lanes 7–10*), as previously shown (*Tian et al., 2016*). This segment of DNA was not precipitated by a control anti-IgG (*lanes 3 and 4*). The BHLHE40, LXRα and C/EBPβ antibodies did not precipitate a segment of DNA that is 4700 bp upstream of the transcription start site, as visualized with primer pair 2 (*lanes 5–10*). Similar results were obtained in two other experiments. These data show that BHLHE40 binds to the SREBP-1c promoter/enhancer region that contains two LXR elements.

We showed previously that the induction of SREBP-1c mRNA by insulin can be blocked by rapamycin, an inhibitor of the mTOR pathway (*Li et al., 2010*; *Owen et al., 2012*). *Kanai et al. (2017)* showed that insulin induction of BHLHE40 in cultured hepatoma cells is also inhibited by rapamycin. To determine whether rapamycin blocks insulin-mediated induction of BHLHE40 in vivo, we subjected rats to fasting or refeeding for 6 hr. At the time of refeeding, the rats were injected intraperitoneally with vehicle or vehicle containing rapamycin. Refeeding increased the SREBP-1c mRNA by 26-fold and this was reduced to 4.4-fold by rapamycin (*Figure 5A*). In vehicle-injected rats, the

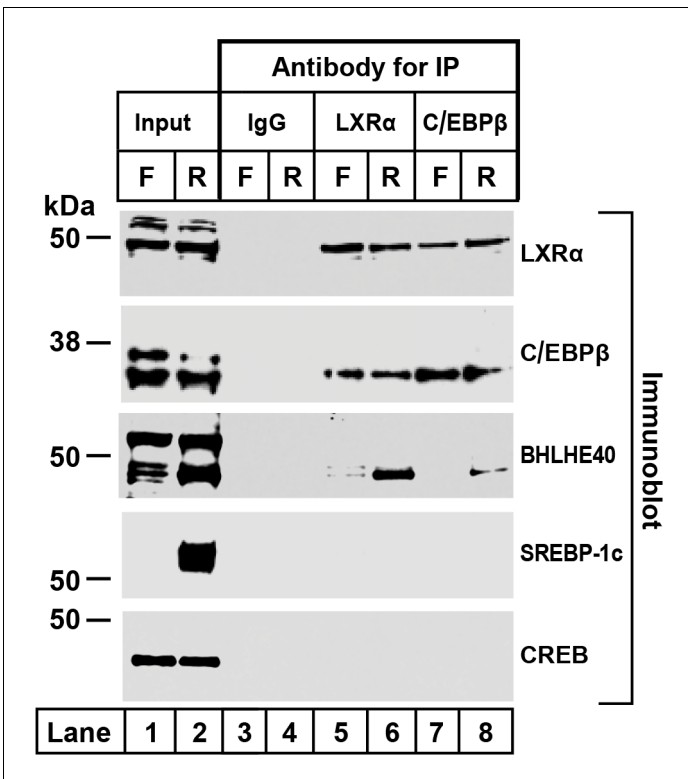

**Figure 3.** Co-immunoprecipitation of LXRα, C/EBPβ, and BHLHE40 in livers from fasted and refed rats. Nuclear extracts were prepared from livers of rats subjected to 48 hr fasting (F) or 48 hr fasting followed by 6 hr refeeding (R). Immunoprecipitations were carried out with agarose beads conjugated with the indicated antibody as described in Materials and methods. After centrifugation, the beads were washed and boiled in SDS loading buffer to elute the proteins. Aliquots of the input (30% of total) and eluates from the immunoprecipitations (100% of eluated fraction) were subjected to SDS-PAGE and immunoblotted with the indicated antibodies as follows: 2 µg/ml of polyclonal anti-LXRα, a 1:1000 dilution of polyclonal anti-C/EBPβ, 1 µg/ml of polyclonal anti-BHLHE40, 1 µg/ml of monoclonal anti-SREBP-1, or a 1:1000 dilution of monoclonal anti-CREB. Proteins were detected with the LI-COR Odessy Infrared Imaging System using a 1:5000 dilution of anti-rabbit IgG conjugated to horseradish peroxidase.

DOI: https://doi.org/10.7554/eLife.36826.004

BHLHE40 mRNA rose by 7.2-fold after refeeding. This increase was reduced to 2.9-fold when the vehicle contained rapamycin (*Figure 5B*).

The finding that rapamycin inhibits insulin induction of BHLHE40 mRNA raised the possibility that rapamycin blocks insulin induction of SREBP-1c mRNA by blocking induction of BHLHE40. To test this hypothesis, we treated fresh rat hepatocytes with or without rapamycin, stimulated them with insulin and then measured the levels of mRNAs encoding BHLHE40 and SREBP-1c (*Figure 6*). In the absence of rapamycin, insulin induced the BHLHE40 mRNA, and this induction was partially blocked by rapamycin. Infection with Lenti-B40, a recombinant lentivirus encoding BHLHE40, raised the level of BHLHE40 mRNA in the absence of insulin and prevented the rapamycin-mediated reduction of BHLHE40 mRNA in the presence of insulin (*Figure 6A*). As shown in *Figure 6B*, insulin induction of SREBP-1c mRNA was inhibited by rapamycin, and this inhibition persisted even when BHLHE40 mRNA was restored with Lenti-B40. These results indicate that mTORC1 plays two independent roles in the insulin-mediated induction of SREBP-1c mRNA, one pertaining to BHLHE40 and the other directly affecting SREBP-1c.

The data of *Figure 6* also indicate that BHLHE40 is not sufficient in itself to induce SREBP-1c mRNA. As shown in *Figure 6A*, in the absence of insulin, infection with Lenti-B40 increased the BHLHE40 mRNA to levels similar to those seen after insulin treatment. However, this increase did not lead to a substantial increase in SREBP-1c mRNA in the absence of insulin (*Figure 6B*).

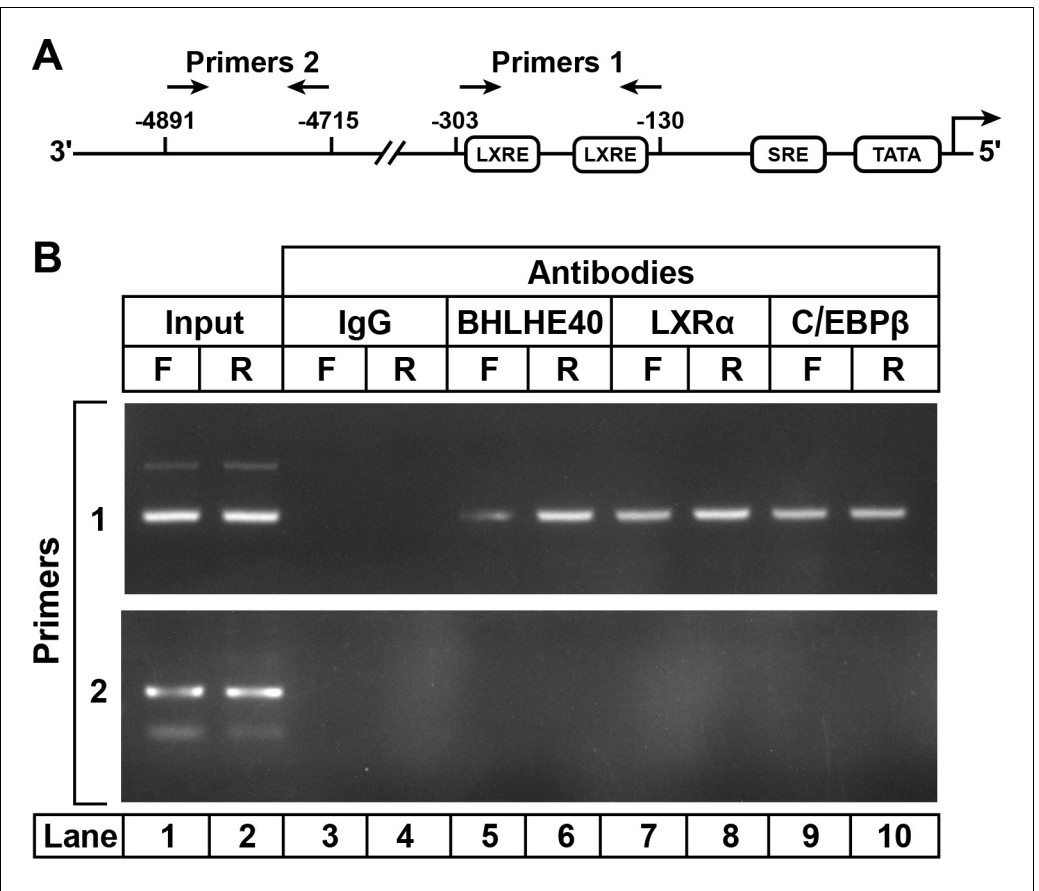

**Figure 4.** ChIP assay to demonstrate BHLHE40 binding to the enhancer/promoter region of rat SREBP-1c gene. (A) Schematic diagram of rat SREBP-1c enhancer/promoter region, showing the location of the two LXREs. Arrows denote the location of the primers used in the ChIP assays. (B) ChIP assay was performed with livers from rats that had been subjected to either 48 hr fasting (F) or 48 hr fasting followed by 6 hr refeeding (R), as described in Materials and methods. After crosslinking, the sheared chromatin was incubated with 5 µg/mL of one of the following antibodies: anti-IgG (control), anti-BHLHE40, anti-LXRα, or anti-C/EBPβ, after which the DNA was purified from each immunoprecipitate and subjected to PCR with primer pairs 1 or 2, and the products visualized on an agarose gel. The DNA sequences for primers 1 and 2 are described in Table 2 in the paper by *Tian et al. (2016)*.

DOI: https://doi.org/10.7554/eLife.36826.005

If the insulin-mediated induction of BHLHE40 is a prerequisite for the induction of SREBP-1c, then the increase in BHLHE40 mRNA must occur before the increase in SREBP-1c mRNA. To test this hypothesis, we measured the amounts of these mRNAs in rat livers at various times after refeeding (*Figure 7A–C*). The BHLHE40 mRNA rose nearly threefold within 1 hr, which was the earliest time point tested (*Figure 7A*). The SREBP-1c mRNA was still low after 2 hr and rose dramatically thereafter (*Figure 7B*). *Figure 7C* compares the relative increases in the BHLHE40 and SREBP-1c mRNAs, showing clearly that the rise in BHLHE40 mRNA precedes the increase in SREBP-1c mRNA. We also conducted a time course study in primary rat hepatocytes (*Figure 7D–F*). The BHLHE40 mRNA rose within 1 hr after addition of insulin to the hepatocytes (*Figure 7D*), and this preceded the increase in SREBP-1c mRNA (*Figure 7E*). *Figure 7F* shows an experiment in which we measured mRNAs in hepatocytes at shorter times after adding insulin. The BHLHE40 mRNA rose within 30 min after adding insulin. The speed of the increase was similar to the fall in PEPCK mRNA, which is known to respond rapidly to insulin (*Granner et al., 1983*). In contrast, the SREBP-1c mRNA did not increase even at 1 hr. These findings identify BHLHE40 as an early response gene to insulin.

To confirm that BHLHE40 is required for the insulin-mediated induction of SREBP-1c mRNA, we treated primary rat hepatocytes with two different siRNAs targeting the BHLHE40 mRNA and then

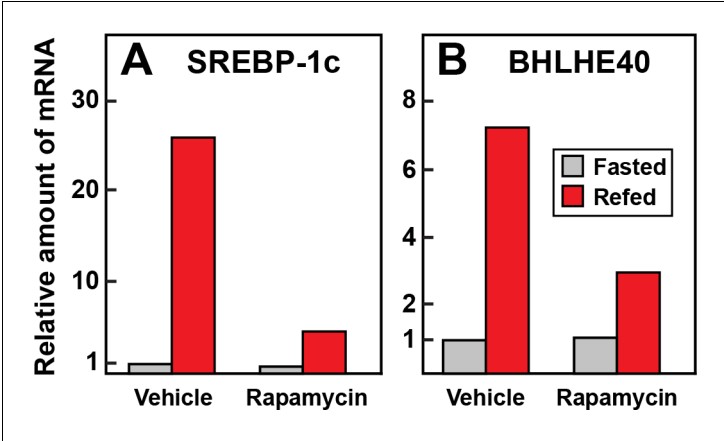

**Figure 5.** Requirement for mTOR in induction in mRNAs encoding SREBP-1c (**A**) and BHLHE40 (**B**) in livers of rats subjected to fasting and refeeding. Sixteen male rats (age 2–3 months) were fasted for 48 hr. Six hours prior to sacrifice, eight rats received an intraperitoneal injection of 20 mg/kg rapamycin, and the other eight received vehicle. Four of the animals in the treated and untreated groups continued to fast, and the other four rats were refed a high-carbohydrate diet as described in Materials and methods. After 6 hr, the rats were sacrificed, and livers were homogenized. Equal amounts of total RNA from the livers of the four rats in each group were pooled and subjected to quantitative RT-PCR. Each bar represents the amount of mRNA relative to that of the vehicle-treated fasted group, which was defined as 1.0. Ct values for BHLHE40 and SREBP-1c in the fasted and vehicle treated group were 24.0 and 23.7, respectively.
DOI: https://doi.org/10.7554/eLife.36826.006

added insulin (*Figure 8*). Both siRNAs (designated siB40A and siB40B) reduced the basal and insulin-stimulated level of BHLHE40 mRNA by about 60% (*Figure 8A*). The two siRNAs also reduced the basal level of SREBP-1c mRNA and diminished the response to insulin (*Figure 8B*). It should be noted that the fivefold stimulation of SREBP-1c mRNA in this transfection experiment is smaller than the 20-fold stimulation in the experiment of *Figure 7E*, which did not include transfection. We

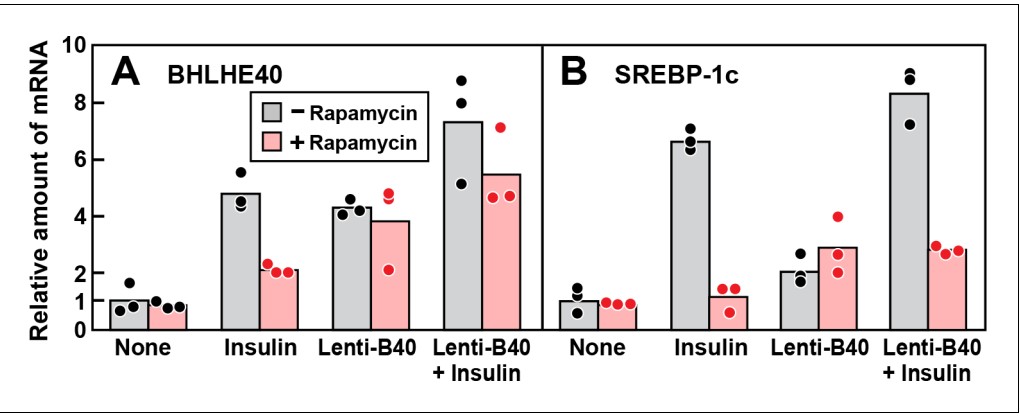

**Figure 6.** Overexpression of BHLHE40 fails to overcome rapamycin-mediated inhibition of SREBP-1c induction by insulin in primary rat hepatocytes. Rat hepatocytes were prepared and plated on day 0 as described in Materials and methods. At 2 hr after plating, cells were infected with or without lentivirus expressing rat *BHLHE40* (Lenti-B40; $5 \times 10^5$ TU/well) as indicated. On day 1, cells were pretreated with or without 100 nM rapamycin for 30 min, after which the cells received either no insulin or 100 nM insulin for 6 hr as indicated. The cells were then harvested, and the levels of mRNA encoding BHLHE40 (**A**) and SREBP-1c (**B**) were determined by RT-PCR. Each dot represents a value from an individual dish. Mean Ct values for BHLHE40 and SREBP-1c in the untreated control groups were 22.8 and 25.6, respectively.
DOI: https://doi.org/10.7554/eLife.36826.007

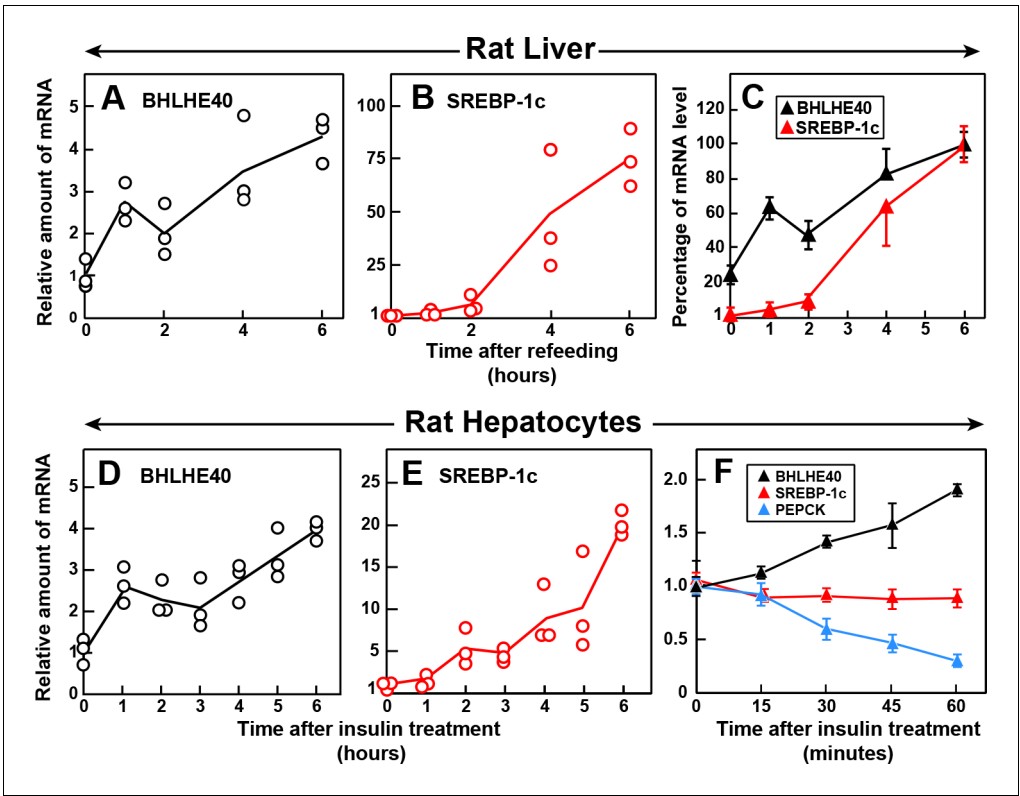

**Figure 7.** Time course of induction of BHLHE40 and SREBP-1c mRNA in livers of refed rats (**A–C**) and in primary rat hepatocytes treated with insulin (**D–F**). (**A–C**) Male rats (age, 2–3 months) were fasted for 48 hr and then refed with a high-carbohydrate diet for the indicated time, after which total RNA from the liver was subjected to quantitative RT-PCR. Each point in (**A**) and (**B**) represents the mRNA level from a single animal relative to the mean value from three fasted rats (i.e. zero-time values). Zero-time Ct values for BHLHE40 and SREBP-1c were 23.5 and 25.7, respectively. Values in (**C**) represent the mean ± SEM of the values from (**A**) and (**B**) plotted as the percentages of the 6 hr value, which is defined as 100%. (**D–F**) Hepatocytes from nonfasted male rats (age, 2–3 months) on a chow diet were prepared and plated on day 0. On day 1, the cells received either no insulin or 100 nM insulin for the indicated time, after which the cells were harvested for measurement of total RNAs by quantitative RT-PCR. Each value in (**D**) and (**E**) (6 hr time course) represents the amount of mRNA from a single dish relative to that of the mean value from the three dishes at zero-time, which is defined as 1.0. Mean Ct values (zero-time) for BHLHE40 and SREBP-1c in the absence of insulin were 23.6 and 26.4, respectively. The values in (**F**) (1 hr time course) represent the mean ±SEM of the values from three dishes. Mean Ct values (zero-time) for BHLHE40, SREBP-1c, and PEPCK in the absence of insulin were 23.6, 26.3, and 20.0, respectively.
DOI: https://doi.org/10.7554/eLife.36826.008

attribute this difference to a nonspecific effect of transfection in the primary hepatocytes. Neither of the two siRNAs decreased the expression of LXRα (*Figure 8C*) nor C/EBPβ (*Figure 8D*).

To determine whether BHLHE40 is required in vivo for insulin induction of SREBP-1c, we produced knockout mice derived from embryonic stem cells that harbor a gene trap in the intron between exons 3 and 4 of the *Bhlhe40* gene. This gene-trap mutation produces a truncated transcript that eliminates exons 4 and 5, which encode amino acids 86–411 of the 411-amino acid protein. Knockout mice homozygous for this mutation grew normally and had normal body weights. At 8 weeks of age, the body weights of WT and knockout mice were 20.0 and 19.8 g, respectively.

*Figure 9A* shows an immunoblot of nuclear extracts from the livers of wild type (WT) and *Bhlhe40*[-/-] (KO) mice. As compared to WT liver, the KO liver lacks the 45.4 kDa BHLHE40 protein (*lanes 1 and 2*). Several bands in the 75–150 kDa range were observed in both the WT and KO samples, one of which was reduced in the KO liver. We believe that these are unrelated proteins recognized nonspecifically by the polyclonal anti-BHLHE40 antibody. *Figure 9B and C* show an experiment in which WT and *Bhlhe40* KO mice were fasted overnight and then refed a high-

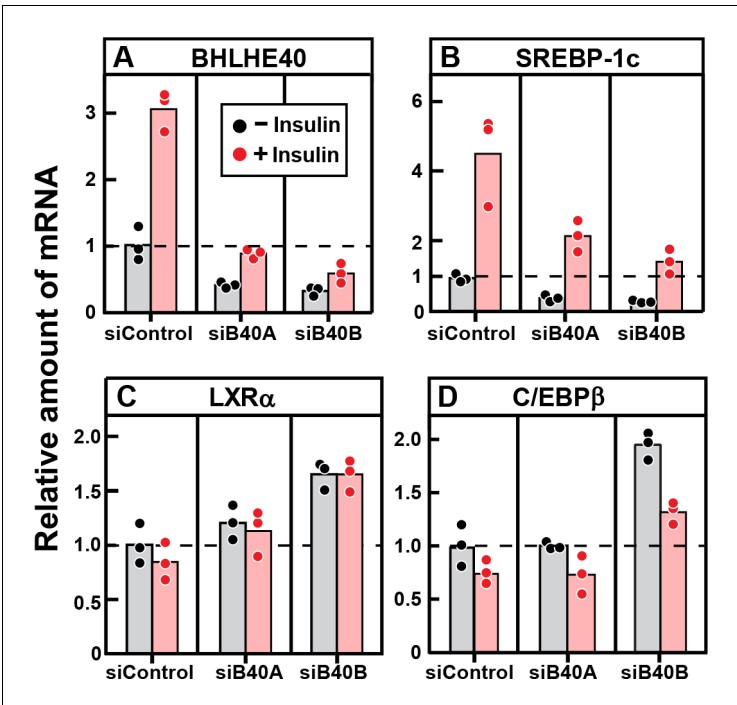

**Figure 8.** Knockdown of BHLHE40 mRNA in rat hepatocytes reduces insulin-induced mRNA for SREBP-1c, but not mRNA for LXRα or C/EBPβ. Hepatocytes from non-fasted male rats (age, 2–3 months) on a chow diet were prepared and plated on day 0. After attachment for 2 hr, cells were transfected with either 50 nM control siRNA (siControl) or one of two siRNAs targeting BHLHE40 (siB40-A or siB40-B). On day 1, cells were treated with or without 100 nM insulin for 6 hr and then harvested for measurement of the indicated mRNA by quantitative RT-PCR. Each value represents an individual dish. Mean Ct values in the absence of insulin for BHLHE40, SREBP-1c, LXRα, and C/EBPβ were 21.8, 24.7, 25.5, and 22.2, respectively.

DOI: https://doi.org/10.7554/eLife.36826.009

carbohydrate diet for 4 hr. Hepatic mRNA levels were measured separately for male (*Figure 9B*) and female (*Figure 9C*) mice. In both genders, the BHLHE40 mRNA was undetectable in the livers from refed knockout mice, and the SREBP-1c mRNA was reduced dramatically. On the other hand, the knockout produced no significant change in the mRNAs encoding LXRα or C/EBPβ.

To confirm that the reduction in SREBP-1c mRNA in the knockout mice was caused by the loss of BHLHE40, we injected the mice with a recombinant adeno-associated virus (AAV) encoding BHLHE40 or a nonspecific control protein. To determine whether phosphorylation at serine 383 is essential, we also injected an AAV encoding a mutant form of BHLHE40 in which serine 383 was replaced with alanine. Three weeks after virus injection, the mice were fasted and refed. In the refed knockout mice, BHLHE40 mRNA was undetectable (*Figure 10B*). Both BHLHE40 viruses restored BHLHE40 mRNA to levels that were approximately twofold above the levels seen in WT mice which are normalized to 1.0 in *Figure 10A*. In WT mice, the injection of virus encoding WT or BHLHE40 (S383A) raised the level of SREBP-1c mRNA by about 1.5-fold (*Figure 10C*). In the knockout mice, the level of SREBP-1c mRNA was low (*Figure 10D*). The level was increased by eightfold when the mice received virus encoding either WT or BHLHE40(S383A) (*Figure 10D*). The *Bhlhe40* knockout had no effect on levels of mRNAs encoding LXRα (*Figure 10F*) or C/EBPβ (*Figure 10H*), and the viruses encoding BHLHE40 did not alter these levels significantly. A small decrease in LXRα expression (28%) was seen in WT mice receiving AAV-BHLHE40 (*Figure 10E*), the functional significance of which is not known.

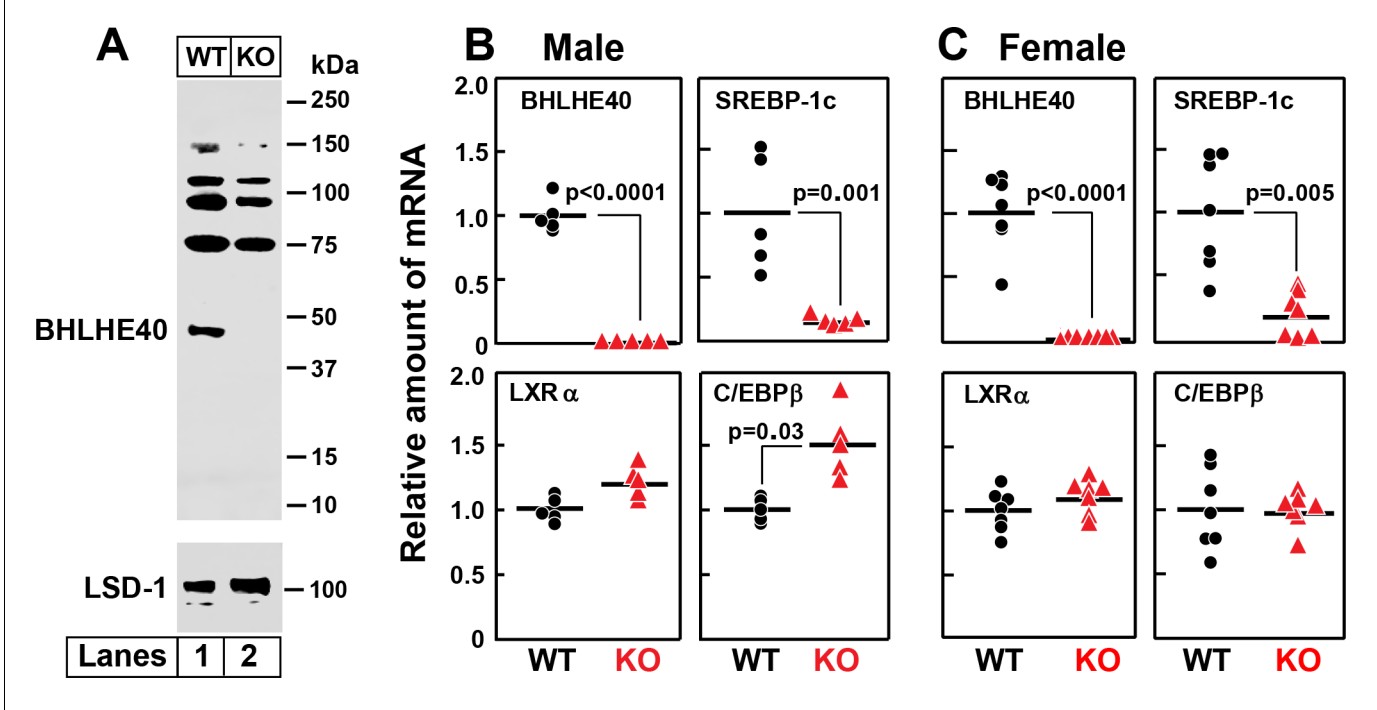

**Figure 9.** Knockout of *Bhlhe40* gene in mice decreases SREBP-1c mRNA in livers of refed animals. (**A**) Immunoblot analysis of liver nuclear extracts from wild type (WT) and *Bhlhe40⁻/⁻* (KO) mice. Nuclear extracts (20 μg protein) were subjected to 4–12% SDS-PAGE, followed by immunoblot analysis with 1.5 μg/ml of rabbit polyclonal anti-BHLHE40 (directed against amino acids 1–60 of rat BHLHE40 protein). LSD1 (lysine-specific demethylase 1) served as a loading control and was immunoblotted with a 1:1000 dilution of rabbit monoclonal anti-LSD1. Proteins were detected with the LI-COR Odyssey Infrared Imaging System using a 1:5000 dilution of anti-rabbit IgG conjugated to horseradish peroxidase. (**B and C**) WT and *Bhlhe40* KO mice were fasted overnight and then refed a high-carbohydrate diet for 4 hr, after which total liver RNA was prepared and subjected to quantitative RT-PCR. Each circle (WT) or triangle (KO) represents an individual mouse. The mean value for WT mice is defined as 1.0. (**B**) mRNAs in livers of male WT and KO mice (age 6 wk; 5 mice/group). Mean Ct values for BHLHE40, SREBP-1c, LXRα and C/EBPβ in WT mice were 21.2, 22.2, 21.7, and 22.1, respectively. (**C**) mRNAs in livers of female WT and KO mice (age 7 wk; 7 mice/group). Mean Ct values for BHLHE40, SREBP-1c, LXRα and C/EBPβ in WT mice were 21.5, 21.0, 22.3, and 22.5, respectively. p-Values calculated using Student t-test.

DOI: https://doi.org/10.7554/eLife.36826.010

## Discussion

The current data fit one more piece into the puzzle of insulin regulation of SREBP-1c mRNA in liver. Solving this puzzle is important clinically because insulin stimulation of SREBP-1c is postulated to underly the fatty liver that occurs in obese people who have high plasma insulin levels (*Brown and Goldstein, 2008*; *Moon et al., 2012*). Previously we showed that two transcription factors are required for insulin induction of SREBP-1c, namely, LXRα and C/EBPβ (*Chen et al., 2004*; *Tian et al., 2016*). Now we add a third factor: BHLHE40, a basic-helix-loop-helix transcription factor that has been studied in many organs and has received multiple names as described in the Introduction.

Our current data show that the level of BHLHE40 mRNA in liver increases rapidly when fasted rats are refed (*Figure 7A*). In freshly isolated rat hepatocytes, insulin increased the amount of BHLHE40 within 30 min (*Figure 7F*). In both cases, the increase in BHLHE40 mRNA preceded the increase in SREBP-1c mRNA. BHLHE40 is essential for the subsequent insulin induction of SREBP-1c mRNA as revealed by experiments in which siRNAs were used to decrease the amount of BHLHE40 in fresh hepatocytes (*Figure 8*). Similarly, the refeeding-induced elevation of SREBP-1c mRNA was severely blunted in livers of *Bhlhe40* knockout mice, and this was reversed when the mice were injected with an adeno-associated virus encoding BHLHE40 (*Figure 10*).

One reason for the relatively slow increase in SREBP-1c mRNA upon refeeding (*Figure 7B and C*) may lie in the necessity to destroy an inhibitor of SREBP-1c transcription. In this regard, *Takeuchi et al. (2016)* reported that KLF15, a ubiquitous transcription factor, forms a complex with LXR that is bound to the SREBP-1c promoter. KLF15 recruits RIP140, a transcriptional co-repressor.

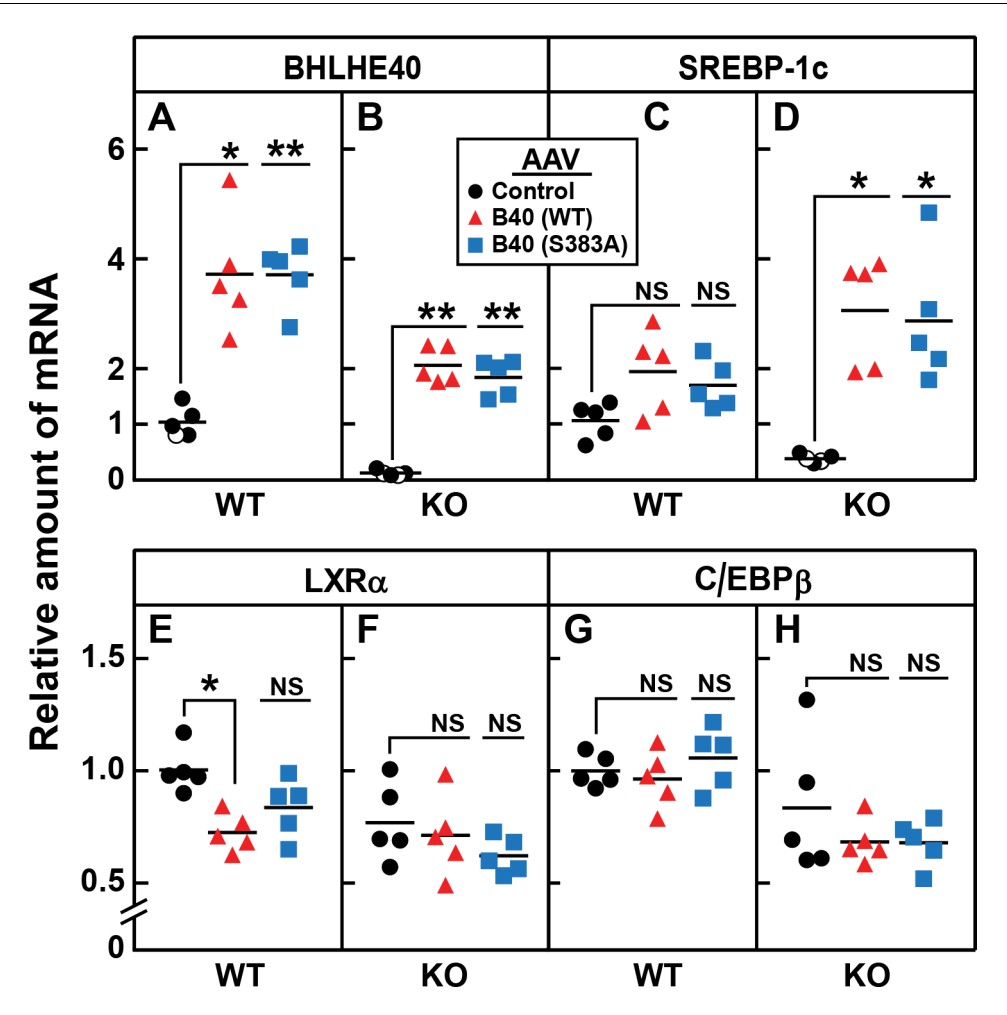

**Figure 10.** AAV-mediated injection of rat BHLHE40 restores the decrease of SREBP-1c mRNA in livers of refed *Bhlhe40* knockout mice. WT and KO mice (age 6 wk; 4 mice/group) received tail vein injections of AAV encoding a control mRNA (AAV-EGFP), wild-type BHLHE40, or mutant BHLHE40(S383A). Three weeks after injection, mice were fasted overnight and then refed with a high-carbohydrate diet for 4 hr, after which total liver RNA was prepared and subjected to quantitative RT-PCR. Each circle, triangle, or square represents an individual mouse. The mRNA expression is plotted as the amount relative to livers of WT mice injected with the control AAV-EGFP, which is assigned a value of 1.0. Mean Ct values for BHLHE40, SREBP-1c, LXRα and C/EBPβ in the livers of WT mice receiving AAV-EGFP were 22.3, 25.4, 22.4, and 22.7, respectively. *p<0.01; **p<0.001; NS, not significant. p-Values calculated using Student t-test.
DOI: https://doi.org/10.7554/eLife.36826.011

The KLF15 gene is activated by cyclic AMP, which is high in fasted liver, owing to the action of glucagon (*Teshigawara et al., 2005*). Upon refeeding, production of KLF15 mRNA is repressed, and KLF15 protein levels decline in concert with the increase in SREBP-1c mRNA (*Takeuchi et al., 2016*). Our laboratory showed previously that glucagon blocks the induction of SREBP-1c mRNA by insulin (*Shimomura et al., 2000*). The data in the current paper raise the possibility that glucagon increases KLF15 protein in livers of fasted rodents and that BHLHE40 cannot activate the SREBP-1c gene until KLF15 has been destroyed.

We identified BHLHE40 in a screen for phosphorylated proteins that were induced in rat liver by refeeding and were precipitated by antibodies against both C/EBPβ and LXRα. Mass spectrometry showed that BHLHE40 is phosphorylated on serine 383 (*Figure 1*). Experiments with fresh hepatocytes demonstrated that the increase in BHLHE40 is mediated by insulin, and it is required for insulin induction of SREBP-1c mRNA. However, the increase in BHLHE40 is not sufficient to induce SREBP-

1c mRNA. Indeed, treatment of fresh hepatocytes with a lentivirus encoding BHLHE40 did not increase the amount of SREBP-1c mRNA in the absence of insulin (*Figure 6B*).

It was attractive to consider that insulin also increases phosphorylation of BHLHE40, and phosphorylation is required to increase transcription of the *SREBP-1c* gene. However, this is not the case as indicated by the finding that insulin responsiveness in BHLHE40-deficient mouse liver was restored by an adeno-associated virus encoding a mutant form of BHLHE40 in which serine 383 was changed to alanine (*Figure 10*). Thus, insulin-mediated induction of SREBP-1c transcription must require yet an additional factor that is activated by insulin. Further studies to identify this missing factor are under way.

# Materials and methods

## Reagents

| Description | Source or reference | Identifier |
|---|---|---|
| 3,3',5-Triiodo-L-thyronine | Sigma-Aldrich, St. Louis, MO | T2877 |
| Bovine insulin | Sigma-Aldrich | I6634 |
| Bio-Gen PRO 200 Homogenizer | PRO Scientific, Oxford, CT | 01–01200 |
| Calpain inhibitor I (ALLN) | AG Scientific, San Diego, CA | CAS110044-82-1 |
| Chromatin Immunoprecipitation (ChIP) Assay Kit | EMD Millipore Corp, Billerica, MA | 17–295 |
| Collagen I-coated dishes | BD Sciences, Franklin Lakes, NJ | 356400 |
| Dexamethasone | Sigma-Aldrich | D4902 |
| DMEM | Sigma-Aldrich | D6046 |
| Freund's Adjuvant, Incomplete | Sigma-Aldrich | F5506 |
| Halt Phosphatase Inhibitor Cocktail | Thermo Fisher Scientific, Waltham, MA | 78426 |
| High-Carbohydrate/Fat-free Diet | MP Biomedicals, Santa Ana, CA | 960238 |
| Lentiviurs Encoding Rat *BHLHE40* | Origene, Rockville, MD | BR206868LV |
| Lipofectamine 2000 | Invitrogen, Carlsbad, CA | 11668–027 |
| Medium 199 | Invitrogen | 11150–059 |
| nProtein A Sepharose 4 Fast Flow Affinity Media | GE Healthcare, Chicago, IL | 17-5280-04 |
| PBS | Sigma-Aldrich | D8537 |
| Phenylmethysulfonyl fluoride (PMSF) | Sigma-Aldrich | P7626 |
| Pierce Co-Immuno- precipitation Kit | Thermo Fisher Scientific | 26149 |
| Protease Inhibitor cOmplete Tablets | Roche Holding AG, Basel, CH | 5892791001 |
| Proteinase K | Thermo Fisher Scientific | AM2546 |
| ProteoSilver Stain Kit | Sigma-Aldrich | PROTSIL1 |
| Rapamycin | Sigma-Aldrich | R0395 |
| Teklad Global 18% Protein Rodent Diet (chow diet for mice) | Harlan Laboratories, Indianapolis, IN | 2018 |
| Teklad Global 16% Protein Rodent Diet (chow diet for rats) | Harlan Laboratories | 2016 |

## Antibodies

| Description | Source or reference | Identifier |
| --- | --- | --- |
| Monoclonal mouse anti-mouse C/EBPβ | Thermo Fisher Scientific | MA1-827 |
| Monoclonal mouse anti-IgG1 isotype control | Cell Signaling Technology, Danvers, MA | 5415 |
| Monoclonal mouse anti-rat LXRα | *Tian et al. (2016)* | Clone 2B7 |
| Monoclonal rabbit anti-human CREB | Cell Signaling Technology | 9197 |
| Monoclonal rabbit anti-mouse SREBP1 | *Rong et al., 2017* | Clone 20B12 |
| Monoclonal rabbit anti-human LSD1 (lysine-specific demethylase 1) | Cell Signaling Technology | 2184 |
| Polyclonal rabbit anti-mouse C/EBPβ | Cell Signaling Technology | 3087 |
| Polyclonal rabbit anti-rat BHLHE40 | This manuscript | IgG-665D |
| Polyclonal rabbit anti-rat LXRα | *Tian et al. (2016)* | IgG-651B |
| Rabbit IgG HRP Linked Whole Ab | GE Healthcare | NA934 |

## Animals

| Description | Source of reference | Identifier |
| --- | --- | --- |
| Sprague-Dawley Rats | Harlan Laboratories, Indianapolis, IN | Order no. 002 |
| NewZealand White Rabbits | Charles River Laboratories, Wilmington, MA | Strain code: 052 (CR) |

## Mouse genetic nomenclature

Throughout the manuscript, we have used the more familiar gene names instead of the less familiar mouse designations. For example *Srebp-1c* stands for the *Srebf1*; LXRα stands for *Nr1h3*; C/EBPβ stands for *Cebpb*. For the mRNA designations, we have used the names of the protein encoded by their respective mRNAs.

## Rapamycin and insulin solutions

For cell culture studies, rapamycin was prepared in dimethyl sulfoxide (DMSO) and stored in multiple aliquots at −20°C. For whole animal studies, a stock solution of rapamycin (50 mg/ml) was prepared in 100% ethanol and stored in aliquots at −20°C. Stock solutions of 0.1 mM insulin were prepared in distilled water adjusted to pH 4.5 with glacial acetic acid, stored in aliquots at 4°C, and used within 3 months.

## Generation of BHLHE40 polyclonal antibody

Rabbit polyclonal anti-BHLHE40 was generated by immunizing a New Zealand White rabbit with a bacterially produced GST-fusion protein containing amino acids 1–60 of rat BHLHE40. An initial subcutaneous injection of 500 µg followed by eight boosts of 250 µg were given biweekly in Incomplete Freund's Adjuvant. A final intraperitoneal injection of 250 µg in saline was given as the final boost. The IgG fraction (designated IgG-665D) of the immune serum was prepared by affinity chromatography on a Protein A Sepharose 4 Fast Flow column (GE Healthcare).

## Animals

Animal work described in this manuscript has been approved and conducted under the oversight of the UT Southwestern Institutional Animal Care and Use Committee. All mice were housed in colony cages with a 12 hr light/12 hr dark cycle (dark cycle from 9 pm to 9 am). Mice were fed a chow diet (Teklad Global 18% Protein Rodent Diet 2018). For fasting/refeeding experiments, mice were fasted overnight and then refed at 9 am for the indicated time with a high-carbohydrate/fat-free diet.

Male Sprague-Dawley rats were housed in animal colony cages and maintained on a reverse 12 hr light/12 hr dark cycle (dark cycle from 10 am to 10 pm) and fed a chow diet (Teklad Global 16% Protein Rodent Diet 2016). For fasting/refeeding experiments, the fasted group was fasted for 48 hr, and the refed group was fasted for 48 hr and then refed with a high carbohydrate/fat-free diet (see above) for 6 hr prior to the study. The starting times for the experiments were staggered so that all rats were sacrificed at the same time, which was at 4 pm. For rapamycin-injection experiments, 6 hr prior to sacrifice rats were injected intraperitoneally with 1.2–1.4 ml of either vehicle [14% (v/v) ethanol, 5% (v/v) Tween 80, and 5% (v/v) polyethylene glycol 400] or vehicle containing rapamycin at a dose of 20 mg/kg.

## Transcription factor co-immunoprecipitation (Co-IP)

Isolated livers from fasted and refed rats were homogenized at 4°C in buffer A [10 mM Hepes, pH 7.6, 25 mM KC1, 1 mM sodium EDTA, 2 M sucrose, 10% (v/v) glycerol, 0.15 mM spermine, 2 mM spermidine, 1 mM PMSF, 1 mM DTT, 0.5 mM Pefabloc, 10 µg/ml leupeptin, 5 µg/ml pepstatin, 25 µg/ml ALLN, and 10 µg/ml aprotinin at a ratio of 1 ml of buffer per g of tissue]. Homogenization was carried out with a saw-toothed generator connected to a Bio-Gen PRO200 Homogenizer (PRO Scientific, Oxford, CT; catalog no. 01–01200). The homogenate was laid over a sucrose cushion at the bottom of an ultracentrifuge tube and spun at $1 \times 10^5$ g for 1 hr at 4°C. The resulting nuclear pellet was resuspended in buffer B [10 mM Hepes pH 7.6, 100 mM KC1, 2 mM MgCl$_2$, 1 mM sodium EDTA, 1 mM DTT, 10% glycerol, 1 mM PMSF, 0.5 mM Pefabloc, 10 µg/ml leupeptin, 5 µg/ml pepstatin, 25 µg/ml ALLN, 10 µg/ml aprotinin, one tablet cOmplete per 10 ml, and 1% (v/v) Halt Phosphatase Inhibitor Cocktail]. Ammonium sulfate was added to the resuspended lysate to a final concentration of 0.4 M, after which the mixture was incubated for 45 min on a rotator at 4°C. Each mixture was spun at $3 \times 10^5$ g for 45 min at 4°C, after which the supernatant was transferred to a fresh tube for co-immunoprecipitation experiments.

The co-IP experiments were performed with 5 µg/ml of monoclonal mouse IgG, LXRα antibody IgG-2B7, or polyclonal C/EBPβ antibody using a Pierce co-IP kit from Thermo Scientific. The IgG, LXRα, and C/EBPβ antibody-conjugated resins were prepared according to the manufacturer's instructions. Pooled liver nuclear extracts from four rats (~1 mg protein in 0.1 ml of nuclear extract in buffer B) were then precleared with control agarose resin, added to 50 µl of the antibody-immobilized resin, and incubated in buffer B with gentle end-over-end mixing for 14–16 hr at 4°C. The precipitated pellets were washed with 0.4 ml of buffer B six times (10 min for each wash), eluted with 20 µl of SDS-PAGE sample buffer, and then subjected to SDS-PAGE followed by either immunoblotting or mass spectrometry analysis.

## LC-MS/MS Analysis

Proteins co-immunoprecipitated by both anti-LXRα and anti-C/EBPβ were subjected to SDS-PAGE on 10% gels and stained with ProteoSilver Stain Kit. Each lane was cut into 5–7 slices of roughly equal sizes. The gel pieces were destained and then reduced in 20 mM dithiothreitol at 56°C for 30 min followed by alkylation in 55 mM iodoacetamide in the dark for 1 hr. Protein in the gels were digested in situ with sequence-grade trypsin (Promega, Madison, WI; catalog no. V5111) in 50 mM ammonium bicarbonate at 37°C overnight. Peptides were extracted sequentially with 5% (v/v) formic acid (FA)/50% (v/v) acetonitrile (ACN) and 0.1% FA/75% ACN, vacuum dried, and then resuspended in 0.1% FA.

LC-MS/MS analysis was performed using a Dionex Ultimate 3000 nanoLC system (ThermoFisher Scientific) coupled to a Quadrupole-Orbitrap Hybrid mass spectrometer (Q-Exactive, ThermoFisher Scientific) equipped with a nano-electrospray ion source. Ionization source parameters were set to: positive mode; capillary temperature, 250°C; spray voltage, 2.4 kV. Extracted peptides were fractionated on a homemade analytical column (75 µm ID, 120 mm length) packed with C18 resin (100 Å, 3 µm, MICHROM Bioresources) using a 78 min gradient: 2–30% B in 68 min, 30–35% B in 4 min, 35–40% B in 2 min, 40–60% B in 3 min, and 60–80% B in 1 min (A = 0.1% FA; B = 100% ACN in 0.1% FA). Full scan mass spectra were acquired from m/z 300–1500 with a resolution of 70,000 at m/z = 200 in the Orbitrap. MS/MS spectra (resolution: 17,500 at m/z = 200) were acquired in a data-dependent mode whereby the top 15 most abundant parent ions were subjected to further fragmentation by higher energy collision dissociation (HCD). For protein identification and

**Table 1.** Primer sequence for mRNA measurements
Primer sequences were custom synthesized by Integrated DNA Technologies, Coraville, IA.

| Gene (species) | Sequences for forward and reverse primers (5'–3') |
| --- | --- |
| 36B4 (rat) | Forward: TTCCCACTGGCTGAAAAGGT<br>Reverse: CGCAGCCGCAAATGC |
| Srebp-1c (rat) | Forward: GACGACGGAGCCATGGATT<br>Reverse: GGGAAGTCACTGTCTTGGTTGTT |
| C/Ebpβ (rat) | Forward: AAGCTGAGCGACGAGTACAAGA<br>Reverse: GTCAGCTCCAGCACCTTGTG |
| Lxrα (rat) | Forward: TTCCCACGGATGCTAATGAA<br>Reverse: GAATGGACGCTGCTCAAAGT |
| Bhlhe40 (rat) | Forward: GCTTCCAGGAAACCATTGGA<br>Reverse: GGCTAGGAAGCTGGGCTTCT |
| Apob (mouse) | Forward: CGTGGGCTCCAGCATTCTA<br>Reverse: TCACCAGTCATTTCTGCCTTTG |
| Srebp-1c (mouse) | Forward: GGAGCCATGGATTGCACATT<br>Reverse: GGCCCGGGAAGTCACTGT |
| Bhlhe40 (mouse) | Forward: TGGTGATTTGTCGGGAAGAAA<br>Reverse: ACGGGCACAAGTCTGGAAAC |
| Lxrα (mouse) | Forward: TCTGGAGACGTCACGGAGGTA<br>Reverse: CCCGGTTGTAACTGAAGTCCTT |
| C/Ebpβ (mouse) | Forward: AAGCTGAGCGACGAGTACAAGA<br>Reverse: GTCAGCTCCAGCACCTTGTG |
| Cyclophilin (mouse) | Forward: TGGAGAGCACCAAGACAGACA<br>Reverse: TGCCGGAGTCGACAATGAT |

DOI: https://doi.org/10.7554/eLife.36826.012

phosphorylated peptide detection, the MS/MS spectra were searched using an in-house Mascot server (Matrix Science) against the rat protein database. Cysteine carbamidomethylation was set as a fixed modification, and variable modifications include serine or threonine phosphorylation. To compare the abundance of proteins identified in the fasted and refed groups, we performed label-free quantification using the MaxQuant package, version 1.3.0.5 (*Cox and Mann, 2008*), which incorporates the Andromeda search engine (*Cox et al., 2011*).

## ChIP assay

These assays were performed as previously described (*Tian et al., 2016*) using a Chromatin Immuno-precipitation (ChIP) Assay Kit containing a protein A resin (EMD Millipore Corp, catalog no. 17–295). Portions of male rat liver were sliced into ~0.5-mm-thick fragments. Each slice (70 mg) was incubated with 1% (wt/vol) formaldehyde at room temperature for 10 min to cross-link proteins to DNA. The tissue was sonicated six times for 10 s at 4°C to disrupt the cells and shear the DNA. After centrifugation, the soluble chromatin solution was precleared by precipitation with protein A resin. The supernatant was incubated with the indicated antibody (*Figure 4*) at 4°C for 14–16 hr, after which the mixture was incubated with the protein A resin for 2 hr at 4°C. The precipitated protein-DNA complexes were washed and eluted with the buffers provided by the manufacturer. The eluted DNA was treated with proteinase K at 45°C for 30 min, followed by incubation with 0.2 M NaCl at 65°C for 4 hr, extraction with phenol-chloroform-isoamylalcohol, and ethanol precipitation. The purified DNA was subjected to PCR using the indicated primer pairs (*Figure 4*), followed by electrophoresis on a 3% agarose gel.

## Generation of *Bhlhe40* knockout mice

The *Bhlhe40* knockout mouse strain in the C57BL/6N background was created from an ES cell clone (EPD0208_6_G02, *Bhlhe40*[tm1a(KOMP)Wtsi]) obtained from KOMP Repository (www.komp.org) and generated by the Wellcome Trust Sanger Institute as a part of a 'gene trap' project designed to inactivate mouse genes (*Skarnes et al., 2011*). The ES clone was injected into Albino C57BL/6N

blastocysts by the Transgenic Core Facility at UT Southwestern Medical Center. The resulting F0 chimeric male founders were then bred with wild-type C57BL/6N female mice to obtain F1 heterozygotes. All experiments were carried out with littermate wild-type and *Bhlhe40*[−/−] mice obtained from intercrosses of *Bhlhe40*[+/−] heterozygotes. Mice were genotyped by PCR using tail genomic DNA with two sets of primers. The first set, 5′-cggatcaaagcttctggtttggagg-3′ and 5′-cctgaccttaacttaggggactccg-3′, amplified a 509 bp fragment only from the wild-type *Bhlhe*40 allele. The second set, 5′-gagatggcgcaacgcaattaatg-3′ and 5′-tgcaatttgccaagatacctggtgg-3′, amplified a 396 bp fragment only from the disrupted *Bhlhe*40 allele.

## Primary rat hepatocytes

Nonfasted rats were anesthetized with isoflurane at the end of the light cycle, and primary hepatocytes were isolated by the collagenase method with modifications as described by *Shimomura et al. (1999)*. On day 0, the isolated hepatocytes were plated onto collagen I-coated, six-well plates (1 × 10⁶ cells/3.5 cm well) in medium A [DMEM supplemented with 5% (v/v) fetal calf serum, 100 U/ml sodium penicillin, and 100 µg/ml streptomycin sulfate]. The cells were incubated at 37°C in 5% $CO_2$. After attachment for 2 hr, the cells were washed once with PBS and changed to medium B (Medium 199 supplemented with 100 nM dexamethasone, 100 nM 3,3′,5-triiodo-L-thyronine, 100 units/ml sodium penicillin, and 100 µg/ml streptomycin sulfate). After 20 hr, the cells were treated with or without insulin and then harvested for measurement of mRNA as described in *Figure Legends*.

## siRNA knockdown

Two hours after plating, triplicate wells of rat hepatocytes were washed once with PBS, switched to 2 ml of serum-free medium B, transfected with 50 nM of synthetic double-stranded siRNAs against BHLHE40

[siRNA-A, rArGrCrArUrUrGrArCrArArArCrCrUrArArUrUrGrArUrCAG; and siRNA-B, rArCrCrArArArGrArArCrUrArArArCrUrCrUrUrGrArGrGrGC (OriGene, catalog no. 4390771)] using Lipofectamine 2000, and incubated in 5% $CO_2$ at 37°C. Sixteen hours after transfection, cells were washed once with PBS and switched to serum-free medium B with the indicated reagents, incubated for 6 hr at 37°C, and then harvested for RNA analysis.

## Quantitative real-time PCR

Total RNA was prepared from rat hepatocytes or mouse liver and subjected to real-time PCR analysis. mRNAs for rat acidic ribosomal phosphoprotein 36B4, mouse apolipoprotein B, and mouse cyclophilin served as invariant controls for rat hepatocytes and mouse liver, respectively, as described previously (*Li et al., 2010*). The primer sequences used for PCR are listed in *Table 1*.

## Recombinant adeno-associated virus (AAV)

A cDNA encoding rat BHLHE40 (NM_053328.1) was cloned into pAAVsc-TBG. The AAV-TBG-rBHLHE40 (WT or S383A) and the control AAV-TBG-EGFP were packaged into AAV8 by Vector Development Core, Horae Gene Therapy Center at University of Massachusetts Medical School. Viral particles were purified by serial $CsCl_2$ centrifugation and stored at 5% (v/v) glycerol in PBS at −80°C. AAVs were administered to mice by tail vein injection at a dose of 3 × 10¹¹ gene copies per mouse. The mice were studied 3 weeks after injection.

## Reproducibility

Similar results were obtained when the experiments were repeated on multiple occasions. Two or three independent studies were done for each experiment except for the LC-MS/MS experiment in *Figure 1*, which was done once.

## Acknowledgements

We thank Guosheng Liang for help with the *Bhlhe40* knockout mice; Christina Li, Linda Donnelly, Angela Carroll, and Jeff Cormier for excellent technical assistance; and Lisa Beatty and Ijeoma Dukes for invaluable help with tissue culture.

## Additional information

### Funding

| Funder | Grant reference number | Author |
|---|---|---|
| National Institutes of Health | HL20948 | Joseph L. Goldstein<br>Michael S Brown |
| Welch Foundation | I-1389 | Zhijian J Chen |

The funders had no role in study design, data collection and interpretation, or the decision to submit the work for publication.

### Author contributions

Jing Tian, Designed research, Performed Research, Wrote the paper, Analyzed the data, Approved the final version; Jiaxi Wu, Xiang Chen, Designed, Performed and analyzed data for LC-MS/MS analysis, Approved final version of paper; Tong Guo, Performed research, Approved final draft of paper; Zhijian J Chen, Designed and analyzed LC-MS/MS analysis, Assisted in preparing manuscript, Approved final version of paper; Joseph L Goldstein, Michael S Brown, Designed experiments, Analyzed data, Wrote the paper, Supervised the research

### Author ORCIDs

Xiang Chen http://orcid.org/0000-0002-6608-3211
Zhijian J Chen http://orcid.org/0000-0002-8475-8251
Joseph L Goldstein https://orcid.org/0000-0002-1894-9463

### Ethics

Animal experimentation: Animal work described in this manuscript has been approved and conducted under the oversight of the UT Southwestern Institutional Animal Care and Use Committee (IACUC).

### Decision letter and Author response

Decision letter https://doi.org/10.7554/eLife.36826.015
Author response https://doi.org/10.7554/eLife.36826.016

## Additional files

### Supplementary files

• Transparent reporting form
DOI: https://doi.org/10.7554/eLife.36826.013

### Data availability

All data generated or analysed during this study are included in the manuscript.

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
