## [Decision Letter]

Thank you for submitting your article "BHLHE40, a third transcription factor required for insulin induction of SREBP-1c mRNA in rodent liver" for consideration by *eLife*. Your article has been reviewed by three peer reviewers, including Peter Tontonoz as the Reviewing Editor and Reviewer #1, and the evaluation has been overseen by Philip Cole as the Senior Editor. The reviewers have opted to remain anonymous.

The reviewers have discussed the reviews with one another and the Reviewing Editor has drafted this decision to help you prepare a revised submission.

Summary:

Hepatic SREBP-1c expression is controlled by insulin but mechanistically how insulin regulates SREBP-1c transcription is not completely understood. Previously it has been shown that LXR and C/EBPβ are required but not sufficient for the regulation of SREBP-1c expression by insulin. In this manuscript, Tian et al. provide evidence indicating BHLHE40 (aka DEC1, STRA13) functions as the third transcription factor required for insulin regulation on SREBP-1c. This study expands the understanding of how insulin controls SREBP-1c expression, which is central to the development of fatty liver and therefore of high importance.

Essential revisions:

1) Figure 3 lacks a specificity control. *Bhlhe40* is only expressed in refed animals, so if it bound non-specifically to IgG one would expect the reported results. This experiment should include a non-specific IgG and ideally the authors should also probe for another diet-independent transcription factor or nuclear protein as a control.

2) It would be informative to perform a ChIP-qPCR analysis to show recruitment of BHLHE40 on SREBP-1c promoter in presence/absence of insulin.

3) The experimental design and data presentation in Figure 8 does not allow the reader to compare fasting and refed states, and therefore to what extent insulin-dependent regulation of SREBP-1c occurs in *Bhlhe40* KO animals. Recommend testing SREBP-1c mRNA levels in fasted and refed wildtype and KO animals to permit direct assessment of the regulation.

4) The authors did not investigate the phenotypical effect of BHLHE40 deletion. Are mice protected from the development of fatty liver when they are exposed to a high fat/high carb diet?

5) The authors cite work from 2017 indicating that *Bhlhe40* is regulated by insulin, but do not address other relevant published data. In the Discussion, the authors should reconcile their results with those of Shen et al., 2014 JBC 289:23332-42 in which *Bhlhe40* is shown to negatively regulate SREBP-1c activity and triglyceride synthesis. Indeed, overexpression of *Bhlhe40* in mouse liver reduced hepatic and serum triglycerides in both db/db and ob/ob obesity models.

---

## [Author Response]

Essential revisions:1) Figure 3 lacks a specificity control. Bhlhe40 is only expressed in refed animals, so if it bound non-specifically to IgG one would expect the reported results. This experiment should include a non-specific IgG and ideally the authors should also probe for another diet-independent transcription factor or nuclear protein as a control.

As suggested by the reviewers, we have repeated Figure 3 with the inclusion of three controls: a non-specific IgG and antibodies directed to two basic-HLH transcription factors (SREBP-1c and CREB) that do not bind LXRα and C/EBPβ.

2) It would be informative to perform a ChIP-qPCR analysis to show recruitment of BHLHE40 on SREBP-1c promoter in presence/absence of insulin.

As suggested by the reviewers, we have performed a ChIP assay (see Figure 4) to show that the LXRα-C/EBPβ-BHLHE40 complex binds to the SREBP-1c promoter/enhancer region.

3) The experimental design and data presentation in Figure 8 does not allow the reader to compare fasting and refed states, and therefore to what extent insulin-dependent regulation of SREBP-1c occurs in Bhlhe40 KO animals. Recommend testing SREBP-1c mRNA levels in fasted and refed wildtype and KO animals to permit direct assessment of the regulation.

The experiment of Figure 8 (now Figure 9) was designed to determine whether knockout of the *Bhlhe40* gene lowers the amount of SREBP-1c mRNA in livers of refed mice. The amount of SREBP-1c mRNA in fasted wild-type mice is barely detectable so we do not think that the *Bhlhe40* knockout could reduce it any further (see Figure 2B and Figure 3, lane 1; also see Horton, et. al., 1998; Liang, et.al., 2002; Li, et.al., 2010).

4) The authors did not investigate the phenotypical effect of BHLHE40 deletion. Are mice protected from the development of fatty liver when they are exposed to a high fat/high carb diet?

A comprehensive study on the development of fatty liver in the BHLHE40 mice will be the subject of a follow-up paper.

5) The authors cite work from 2017 indicating that Bhlhe40 is regulated by insulin, but do not address other relevant published data. In the Discussion, the authors should reconcile their results with those of Shen et al., 2014 JBC 289:23332-42 in which Bhlhe40 is shown to negatively regulate SREBP-1c activity and triglyceride synthesis. Indeed, overexpression of Bhlhe40 in mouse liver reduced hepatic and serum triglycerides in both db/db and ob/ob obesity models.

Shen, et al. conclude that Dec1 (BHLHE40) negatively regulates hepatic SREBP-1c expression – a conclusion opposite to our findings. Shen’s results may be explained as follows: 1) all of their experiments were carried out in the fasted state where plasma insulin levels and SREBP-1c expression are extremely low; 2) they did not carry out any experiments to show the relation between Dec1 expression and insulin, i.e., there were no fasting/refeeding experiments and no study of hepatocytes treated with insulin; 3) their conclusions were based largely on adenoviral injections, which unlike AAV injections frequently cause nonspecific inflammatory effects in the liver. Inasmuch as this study is not comparable to ours, we have chosen not to cite it.